# A Verifiable Arbitrated Quantum Signature Scheme Based on Controlled Quantum Teleportation

**DOI:** 10.3390/e24010111

**Published:** 2022-01-11

**Authors:** Dianjun Lu, Zhihui Li, Jing Yu, Zhaowei Han

**Affiliations:** 1School of Mathematics and Statistics, Shaanxi Normal University, Xi’an 710119, China; ldj@qhnu.edu.cn (D.L.); 2008011@qhnu.edu.cn (J.Y.); hanzw888@snnu.edu.cn (Z.H.); 2School of Mathematics and Statistics, Qinghai Normal University, Xining 810008, China

**Keywords:** arbitrated quantum signature, controlled quantum teleportation, von Neumann measurement, Bell measurement, verifiability

## Abstract

In this paper, we present a verifiable arbitrated quantum signature scheme based on controlled quantum teleportation. The five-qubit entangled state functions as a quantum channel. The proposed scheme uses mutually unbiased bases particles as decoy particles and performs unitary operations on these decoy particles, applying the functional values of symmetric bivariate polynomial. As such, eavesdropping detection and identity authentication can both be executed. The security analysis shows that our scheme can neither be disavowed by the signatory nor denied by the verifier, and it cannot be forged by any malicious attacker.

## 1. Introduction

Since Bennett and Brassard [1] proposed the quantum key distribution (QKD) protocol in 1984, quantum cryptography has attracted extensive attention. Its security is guaranteed by the principles of quantum mechanics such as the Heisenberg uncertainty principle and the quantum no-cloning theorem. Quantum cryptography can provide the advantage of unconditional security, making the research of quantum cryptography increasingly important. Many important quantum cryptography branches have been developed, such as quantum key distribution [2,3], quantum signature (QS) [4,5,6], quantum teleportation (QT) [7], quantum authentication [8], and deterministic secure quantum communication [9].

Quantum signatures can be applied to verify the identity of the sender and the integrity of the information. The arbitrated quantum signature (AQS), providing many merits, has attracted much attention. In 2002, Zeng et al. [10] proposed the first arbitrated quantum signature scheme using the Green–Horne–Zeilinger (GHZ) state and the quantum one-time pad (QOTP). Based on the design of the classical arbitrated digital signature, the scheme provides a re-verification service for signatory and receiver using the online signature provided by a trusted third party arbitrator. In 2008, Curty and Lutkenhaus [11] investigated the scheme [10], and they claimed that it was not clearly described and that the safety analysis was incorrect. In response to the controversy of Curty et al., Zeng et al. [12] proved the scheme [10] in more detail. In 2009, to reduce the complexity and improve the efficiency of the protocol [10], Li et al. [13] proposed an AQS scheme based on the Bell states rather than the GHZ states and proved its advantages in terms of transmission efficiency and low complexity. Unfortunately, in 2010, Zou and Qiu [14] argued that Li’s AQS scheme can be disavowed by the receiver, and they proposed an AQS protocol that uses bulletin boards and other security schemes that do not use entangled state. Their scheme further simplified the protocol of Li et al., and an improved AQS scheme was designed using single particles that can resist the denunciation of the receiver, thus reducing the difficulty of the physical implementation of AQS. However, in 2011, Gao et al. [15] conducted the first comprehensive cryptanalysis of previous AQS schemes in terms of forgery and disavowal. They found that the existing AQS schemes based on QOTP encryption [13,14] all have some security problems. In other words, the receiver Bob can realize the existence of the forgery of a signature under the known message attack, while the sender Alice can successfully disavow any signature of hers through a simple attack. Choi et al. [16] found that most AQS protocols can be cracked through a specific existential forgery attack due to the careless taking advantage of the optimal quantum one-time pad based on Pauli operators. To overcome this weakness, they proposed a simple method to ensure the security of the signature. As Choi et al.proved, Bob could not simultaneously forge both the information and the signature to be verified by an arbitrator in the event of a dispute. In the same year, Yang et al. [17] demonstrated how to construct an arbitrated quantum signature protocol for classical messages using untrusted arbitrators. In order to solve the security problems experienced with the AQS protocol, Zhang et al. [18] analyzed the existing security problems [15,16] in 2013 and suggested some corresponding improvement strategies to counter forgery attacks. In order to solve the problem proposed by Gao et al. [15], Liu et al. [19] designed a new QOTP algorithm in 2014, which mainly relies on inserting decoy states into fixed positions, and constructed an unconditionally secure AQS scheme with fast signing and verifying using only a single particle state. In 2015, Li [20] used chained CNOT operation for encryption, instead of quantum one-time pad, to ensure the security of the protocol. To improve the efficiency of quantum bit to 100%, Yang [21] proposed an AQS scheme with the cluster state in 2016. In 2017, in order to resist forgery attacks and disavowal attacks, Zhang et al. [22] proposed a new quantum encryption based on the key-controlled chained CNOT operations (KCCC encryption), and through KCCC encryption, constructed an improved arbitrated quantum signature protocol. In 2016, Yang et al. [23] also proposed a theoretically extensible quantum digital signature with a star-like cluster state. In 2018, Shi et al. [24] proposed an arbitrated quantum signature scheme with the Hamiltonian algorithm based on blind quantum computation. Due to the application of blind quantum computation, it is not necessary to recover the original message during verification, which can improve the simplicity and operability of AQS. In the same year, Feng et al. [25] constructed an AQS scheme based on continuous variable squeezed vacuum states rather than coherent states to further improve coding efficiency and performance. In 2019, Feng et al. [26] proposed an AQS scheme with quantum walk-based teleportation, which does not require the preparation of entangled particles in advance, making the AQS protocol more flexible and practical. In 2020, Chen et al. [27] proposed an offline arbitrated semi-quantum signature scheme based on four-particle cluster states, in which the classical parties can sign with the assistance of a quantum arbitrator. Different from the typical arbitrated quantum signature schemes, the arbitrator in this protocol acts as a relay station of signature transmission and no longer interferes with the direct authentication of the signature, so that the signature receiver has completed authentication rights. There is no additional direct communication between the signatory and the receiver, which reduces the complexity of transmission. However, the above AQS scheme does not consider authentication between signatory, arbitrator, and verifier.

Quantum teleportation is a technology that uses the entangled state or cluster state to transmit information between two sides of communication. The first scheme of quantum teleportation was proposed by Bennett et al. [28] in 1993. It is a scheme of teleportation through classical channel and an EPR entangled channel. In 1998, Karlsson and Bourennane [29] proposed controlled quantum teleportation. Its basic idea is that the receiver reconstructs the unknown quantum state with the help of the controller. Until now, quantum teleportation has been studied using the GHZ states [30], W. states [31,32], cluster states [33], and other entangled states as quantum channels. In recent years, many quantum signature schemes have used entangled states as quantum channels, and methods were proposed to transmit unknown quantum states of a single particle [34] or double particles [35]. In 2005, Brown et al. [36] developed a computationally feasible entanglement measurement method based on negative bias transposition criterion, and found highly entangled four-qubit states and five-qubit states by searching. In 2008, Muralidharan and Panigrahi [37] investigated the usefulness of the five-qubit state introduced by Brown et al. [36] for quantum information applications such as quantum teleportation. The results show that this state can be used for perfect teleportation of arbitrary single- and two-qubit systems.

In this paper, we construct an arbitrated quantum signature scheme that can verify the identity of participants using five-qubit entangled states as quantum channels and controlled quantum teleportation. The security analysis result shows that our AQS scheme ensures that the signatory Alice cannot disavow, the verifier Bob cannot repudiate, and any illegal attacker can not forge. The proposed scheme uses mutually unbiased bases particles as decoy particles. It applies a pair of function values of symmetric binary polynomials to perform a unitary operation on decoy particles so that eavesdropping detection and identity verification between participants can be performed. In addition, the scheme only needs von Neumann measurement, Bell measurement, and a unitary operation to recover the single-particle qubit state. It replicates message from the signatory Alice to the verifier Bob, which is an attractive advantage for realizing an actual quantum communication network.

The scheme has the following advantages:(1)The mutually unbiased bases particles are used as decoy particles to prevent external adversaries from eavesdropping during transmission;(2)The receiver only needs to ask about the position of the decoy particles without asking what the measurement bases are in the process of eavesdropping detection;(3)The scheme provides the function of identity authentication among participants. It uses a pair of function values of symmetric binary polynomials as parameters of the unitary operation, which is used to act on the decoy particles to verify the identity of participants.

The rest of this article are organized as follows. In Section 2, the concepts of the arbitrated quantum signature, mutually unbiased bases and controlled quantum teleportation are introduced. In Section 3, the detailed process of the proposed protocol is described. In Section 4 and Section 5, the verifiability analysis and safety analysis are conducted, respectively. Finally, a brief conclusion is provided in Section 6.

## 2. Preliminaries

In this section, we first briefly review some notions concerning the arbitrated quantum signature scheme and the definition of mutually unbiased bases, which is presented in [38]. Then, we introduce controlled quantum teleportation, which is used in constructing the arbitrated quantum signature scheme. Finally, an example of controlled quantum teleportation is given.

### 2.1. Some Notions concerning the Arbitrated Quantum Signature

A digital signature scheme is a cryptographic primitive that provides the receiver of a message with assurance about the integrity of the data, and the identity of the sender/signatory. Furthermore, it offers unforgeable and undeniable property. Similarly, the arbitrated signature scheme is a digital signature scheme finished with the help of an arbitrator, who is a disinterested third party trusted to complete a protocol. Here “trusted” means that all people involved in the protocol accept what he says as true and what he does as correct, as well as that he will complete his part of the protocol [14]. The quantum signature is a quantum version of the classical digital signature.

### 2.2. Mutually Unbiased Bases

**Definition** **1**([38])**.**
*We suppose that $A_{1}={{\left\{|\varphi_{i}\rangle \right\}}^{q}}_{i=1}$ and $A_{2}={{\left\{|\psi_{i}\rangle \right\}}^{q}}_{i=1}$ are two sets of standard orthogonal bases, which are defined over a q-dimensional complex space $C^{q}$. We state that $A_{1}$ and $A_{2}$ are mutually unbiased if the following relationship is satisfied: $|\langle \varphi_{i}|\psi_{j}\rangle |=\frac{1}{\sqrt{q}}$.*

If any two sets of standard orthogonal bases $A_{1},A_{2},\cdots ,A_{m}$ in space $C^{q}$ is unbiased, then this set is called an unbiased bases set. Additionally, one can find at most q+1 mutually unbiased bases if *q* is an odd prime number. In particular, the computation basis is expressed as $\left\{|k\rangle |k\in D\right\}$, where $D=\left\{0,1,\ldots ,q-1\right\}$. In addition to the computation basis, the remaining *q* groups of unbiased bases can be expressed as $|\varphi_{l}^{\left(j\right)}\rangle =\frac{1}{\sqrt{q}}\sum_{k=0}^{q-1}\omega^{k\left(l+jk\right)}|k\rangle$, where $\omega =e^{\frac{2\pi i}{q}}$ and j∈D represent the number of the mutually unbiased bases and l∈D list the number of vectors for the given bases. For $j\neq j^{\prime }$ these mutually unbiased bases satisfy the following conditions: $|\langle {\varphi_{l}}^{\left(j\right)}|{\varphi_{l}}^{\left(j^{\prime }\right)}\rangle |=\frac{1}{\sqrt{q}}$.

Letting $X_{q}=\sum_{n=0}^{q-1}\omega^{n}\left|n\rangle \langle n\right|$, we have following operations:$$\begin{array}{c} {X_{q}}^{x}|{\varphi_{l}}^{\left(j\right)}\rangle ={X_{q}}^{x}\frac{1}{\sqrt{q}}\sum_{k=0}^{q-1}\omega^{k\left(l+jk\right)}|k\rangle \\ =\frac{1}{\sqrt{q}}\left(\sum_{n=0}^{q-1}\omega^{xn}\left|n\rangle \langle n\right|\right)\left(\sum_{k=0}^{q-1}\omega^{k\left(l+jk\right)}|k\rangle \right)=\frac{1}{\sqrt{q}}\left(\sum_{k=0}^{q-1}\omega^{k\left(l+x\right)+jk}|k\rangle \right)=|{\varphi_{l+x}}^{\left(j\right)}\rangle . \end{array}$$

For the convenience of expression, ${X_{q}}^{x}$ is denoted as $U_{x}$ which is a unitary operator, that is, $U_{x}|{\varphi_{l}}^{\left(j\right)}\rangle =|{\varphi_{l+x}}^{\left(j\right)}\rangle$. Especially, we have $U_{l}|{\varphi_{0}}^{\left(0\right)}\rangle =|{\varphi_{l}}^{\left(0\right)}\rangle$.

### 2.3. Controlled Quantum Teleportation

Our arbitrated quantum signature scheme is based on controlled quantum teleportation. The five-qubit entangled state can be used to perfect the teleportation of arbitrary single- and two-qubit systems [37], which are suitable for maximum contact teleportation and satisfy the biggest task-oriented definition of entangled state [36]. Due to the above advantages, in this section, we use the five-qubit entangled state as the quantum channel to execute controlled quantum teleportation. The design form is as follows:$${|\xi \rangle }_{12345}=\frac{1}{2}{\left(\left|001\rangle \right|\phi^{-}\rangle +|010\rangle |\psi^{-}\rangle +|100\rangle |\phi^{+}\rangle +|111\rangle |\psi^{+}\rangle \right)}_{12345}.$$
In the form above, $|\phi^{+}\rangle ,|\phi^{-}\rangle ,|\psi^{+}\rangle ,|\psi^{-}\rangle$ represent the four Bell states of two particles, respectively, $|\phi^{\pm }\rangle =\frac{1}{\sqrt{2}}\left(|00\rangle \pm |11\rangle \right)$ and $|\psi^{\pm }\rangle =\frac{1}{\sqrt{2}}\left(|01\rangle \pm |10\rangle \right)$. These states exhibit true multipartite entanglement from both negative bias measurements and von Neumann measurements. Even after tracking one or two qubits from this state, entanglement is maintained in the resulting subsystem, which is therefore highly robust.

In the quantum teleportation process, the participants are Alice, Trent, and Bob. Alice owns particles (*M*, 2, 3), Trent owns particles (1, 4), and Bob owns the particle (5).

The model of controlled quantum teleportation is shown in Figure 1.

The working process of the controlled quantum teleportation is described below:

**Step 1:** Alice performs three-particle von Neumann measurements of the particles (*M*, 2, 3) in her possession. The three-particle von Neumann measurement basis is $\left\{|\chi^{i}\rangle \right\}$$\left(i=1,2,\cdots ,8\right)$, as shown in Table 1.

Suppose Alice carries the information of the quantum state of particle *M* as ${|\gamma \rangle }_{M}={\left(\alpha |0\rangle +\beta |1\rangle \right)}_{M}$, where the coefficients α and β are unknown and satisfy ${\left|\alpha \right|}^{2}+{\left|\beta \right|}^{2}=1$. The combined state of the entire system ${|\Psi \rangle }_{M12345}$ consisting of particles *M* and $\left(1,2,3,4,5\right)$ is given by the formula below.
$$\begin{array}{cc} {|\Psi \rangle }_{M12345} & ={|\gamma \rangle }_{M}⊗{|\xi \rangle }_{12345}={\left(\alpha |0\rangle +\beta |1\rangle \right)}_{M}⊗{|\xi \rangle }_{12345}\\ & ={\left(\alpha |0\rangle +\beta |1\rangle \right)}_{M}⊗\frac{1}{2}{\left(\left|001\rangle \right|\phi^{-}\rangle +|010\rangle |\psi^{-}\rangle +|100\rangle |\phi^{+}\rangle +|111\rangle |\psi^{+}\rangle \right)}_{12345}\\ & ={\left(\alpha |0\rangle +\beta |1\rangle \right)}_{M}⊗\frac{1}{2\sqrt{2}}(|00100\rangle \\ & -|00111\rangle +|01001\rangle -|01010\rangle +|10000\rangle +|10011\rangle +|11101\rangle +{\left|11110\rangle \right)}_{12345}\\ & =\frac{1}{2\sqrt{2}}[\alpha |000100\rangle -\alpha |000111\rangle +\alpha |001001\rangle \\ & -\;\alpha |001010\rangle +\alpha |010000\rangle +\alpha |010011\rangle +\alpha |011101\rangle +\alpha |011110\rangle \\ & +\;\beta |100100\rangle -\beta |100111\rangle +\beta |101001\rangle -\beta |101010\rangle \\ & +\;\beta |110000\rangle +\beta |110011\rangle +\beta |111101\rangle +{\beta |111110\rangle ]}_{12345}. \end{array}$$

**Step 2:** Alice conveys her measurement outcomes to Bob through the classical channel. If Alice uses measurement basis $\left\{|\chi^{i}\rangle \right\}$$\left(i=1,2,\cdots ,8\right)$ to measure ${|\Psi \rangle }_{M12345}$, then ${|\Psi \rangle }_{M12345}$ will collapse into the corresponding states shown in Table 2.

**Step 3:** Trent uses Bell measurement basis $\left\{|\phi^{+}\rangle ,|\phi^{-}\rangle ,|\psi^{+}\rangle ,|\psi^{-}\rangle \right\}$ to perform two-particle measurements on particles (1,4). After Trent measures ${\langle \chi_{M23}^{i}|\Psi \rangle }_{M12345}$ with Bell measurement basis $\left\{|\phi^{+}\rangle ,|\phi^{-}\rangle ,|\psi^{+}\rangle ,|\psi^{-}\rangle \right\}$, ${\langle \chi_{M23}^{i}|\Psi \rangle }_{M12345}$ collapses to the corresponding state shown in Table 3.

**Step 4:** Trent sends his measurement results to Bob through the classical channel.

**Step 5:** Following Trent and Alice’s measurements, Bob performs an appropriate unitary operation $U_{\left(5\right)}$ and successfully reconstructs the original unknown quantum state ${|\gamma \rangle }_{M}$ on the particle (5).

The participants’ measurement outcomes and the unitary operation $U_{\left(5\right)}$ are shown in Table 4, in which MO represents the measurement outcomes and all the Pauli matrices are shown below.
$$\begin{array}{c} I=\left[\begin{array}{cc} 1 & 0\\ 0 & 1 \end{array}\right],\;\sigma_{x}=\left[\begin{array}{cc} 0 & 1\\ 1 & 0 \end{array}\right],\;\sigma_{y}=\left[\begin{array}{cc} 0 & -i\\ i & 0 \end{array}\right],\;\sigma_{z}=\left[\begin{array}{cc} 1 & 0\\ 0 & -1 \end{array}\right].\; \end{array}$$

Based on Alice and Trent’s measurement outcomes, Bod performs the corresponding unitary operation $U_{\left(5\right)}$ on particle (5) and his result is α|0〉+β|1〉. This is the original information particle state. That is, Alice successfully transmits the unknown quantum state to Bob under Trent’s control.

**Example** **1.**
*Suppose that the information particle states are {|0〉,|1〉,|+〉,|−〉,|+〉,|0〉,|−〉, |1〉, |1〉,|0〉}. Alice combines each information particle state and five-particle entangled state into a six-particle state sequence: $\{|0\rangle ⊗{|\xi \rangle }_{12345}$, $|1\rangle ⊗{|\xi \rangle }_{12345}$, ${|+\rangle ⊗|\xi \rangle }_{12345}{,|-\rangle ⊗|\xi \rangle }_{12345}{,|+\rangle ⊗|\xi \rangle }_{12345}$, $|0\rangle ⊗{|\xi \rangle }_{12345}{,|-\rangle ⊗|\xi \rangle }_{12345},|1\rangle ⊗{|\xi \rangle }_{12345}$, $|1\rangle ⊗{|\xi \rangle }_{12345}$, $|0\rangle ⊗{|\xi \rangle }_{12345}\}$. Alice performs von Neumann measurement of the particles (M,2,3) in the sequence. Suppose that von Neumann measurement outcomes are $\{\chi^{1},\chi^{5},\chi^{7},\chi^{2},\chi^{8}$, $\chi^{3}$, $\chi^{4},\chi^{4},\chi^{8},\chi^{6}\}$, and Trent’s measurement outcomes of the particles (1,4) in the sequence are $\{|\phi_{14}^{+}\rangle$, $|\phi_{14}^{-}\rangle$, $|\psi_{14}^{+}\rangle$, $|\psi_{14}^{-}\rangle$, $|\phi_{14}^{+}\rangle$, $|\psi_{14}^{+}\rangle$, $|\psi_{14}^{-}\rangle$, $|\phi_{14}^{-}\rangle$, $|\psi_{14}^{+}\rangle$, $|\psi_{14}^{-}\rangle \}$. At this time, the states of all particles (5) should be ${\left(\alpha |1\rangle +\beta |0\rangle \right)}_{5}$, ${\left(\alpha |1\rangle +\beta |0\rangle \right)}_{5}$, ${\left(\alpha |1\rangle +\beta |0\rangle \right)}_{5}$, ${\left(-\alpha |0\rangle +\beta |1\rangle \right)}_{5}$, ${\left(-\alpha |0\rangle +\beta |1\rangle \right)}_{5}$, ${\left(-\alpha |1\rangle -\beta |0\rangle \right)}_{5}$, ${\left(-\alpha |1\rangle +\beta |0\rangle \right)}_{5}$, ${\left(\alpha |0\rangle -\beta |1\rangle \right)}_{5}$, ${\left(-\alpha |1\rangle +\beta |0\rangle \right)}_{5}$, ${\left(-\alpha |0\rangle +\beta |1\rangle \right)}_{5}$. After Bob performs the following unitary operation: $\{\sigma_{x},\sigma_{x},\sigma_{x},-\sigma_{z},-\sigma_{z},-\sigma_{x},-i\sigma_{y},\sigma_{z},-i\sigma_{y},$ $-\sigma_{z}\}$, the states of the information particles are {|0〉,|1〉,|+〉,|−〉,|+〉,|0〉,|−〉,|1〉,|1〉,|0〉}.*


## 3. The Proposed Verifiable Arbitrated Quantum Signature Scheme

In our scheme, Alice the signatory, Bob the verifier, and Trent the arbitrator are defined as the three participants. The arbitrator Trent should be trusted by both Alice and Bob. The detailed procedures of our scheme can be described as follows.

### 3.1. Initializing Phase

**Step I1:** Alice and Trent share secret key $K_{A}$ and Bob and Trent share secret key $K_{B}$. The secret key distribution task can be performed using the QKD protocol, which has been proven to provide unconditional security [39,40].

**Step I2:** Trent selects a k−1-order symmetric binary polynomial: $F\left(x,y\right)=a_{00}+a_{10}x+a_{01}y+a_{11}xy+a_{20}x^{2}+a_{02}y^{2}+a_{12}xy^{2}+a_{21}x^{2}y+a_{22}x^{2}y^{2}+\cdots +a_{k-1,k-1}x^{k-1}y^{k-1}$ mod *q*, where *q* is a prime number, F(x,y)∈GF(q)[x,y], $a_{ij}\in F_{q}$, i,j∈{0,1,⋯,k−1}, $a_{ij}=a_{ji}$, $F_{q}$ is a finite field. Suppose that the public identity information for the participants Alice, Bob, and Trent is $x_{A},x_{B},x_{T}$. Trent computes two share polynomials $f_{A}\left(y\right)=F\left(x_{A},y\right)$ and $f_{B}\left(y\right)=F\left(x_{B},y\right)$. The share polynomial $f_{A}\left(y\right)$ is encrypted as $f_{A}^{{}^{\prime }}\left(y\right)=E_{K_{A}}\left(f_{A}\left(y\right)\right)$ and $f_{A}^{{}^{\prime }}\left(y\right)$ is sent to Alice. The share polynomial $f_{B}\left(y\right)$ is encrypted as $f_{B}^{{}^{\prime }}\left(y\right)=E_{K_{B}}\left(f_{B}\left(y\right)\right)$ and $f_{B}^{{}^{\prime }}\left(y\right)$ is sent to Bob.

**Step I3:** Alice receives $f_{A}^{{}^{\prime }}\left(y\right)$ and decrypts it with secret key $K_{A}$ to obtain $f_{A}\left(y\right)=F\left(x_{A},y\right)$. Alice calculates $f_{A}\left(x_{B}\right)=F\left(x_{A},x_{B}\right)$ and $f_{A}\left(x_{T}\right)=F\left(x_{A},x_{T}\right)$ based on Bob’s and Trent’s public identity information $x_{B}$ and $x_{T}$. Similarly, Bob can calculate $f_{B}\left(x_{A}\right)=F\left(x_{B},x_{A}\right)$ and $f_{B}\left(x_{T}\right)=F\left(x_{B},x_{T}\right)$ based on Alice’s and Trent’s public identity information $x_{A}$ and $x_{T}$. Due to the symmetry of the binary polynomial, $f_{A}\left(x_{B}\right)=f_{B}\left(x_{A}\right)$, $f_{A}\left(x_{T}\right)=f_{T}\left(x_{A}\right)$, $f_{B}\left(x_{T}\right)=f_{T}\left(x_{B}\right)$.

**Step I4:** According to the value of $F(x_{A},x_{B})$ and $F(x_{A},x_{T})$, Alice executes the unitary operations $U_{F(x_{A},x_{B})}$ and $U_{F(x_{A},x_{T})}$ on $\left|\mu \rangle =\right|\varphi_{0}^{\left(0\right)}\rangle =\frac{1}{\sqrt{q}}\sum_{i=0}^{q-1}|i\rangle$ to produce enough decoy particles: ${|\mu \rangle }_{A,B}=U_{F\left(x_{A},x_{B}\right)}|\varphi_{0}^{\left(0\right)}\rangle =|\varphi_{F\left(x_{A},x_{B}\right)}^{\left(0\right)}\rangle$ and ${|\mu \rangle }_{A,T}=U_{F\left(x_{A},x_{T}\right)}|\varphi_{0}^{\left(0\right)}\rangle =|\varphi_{F\left(x_{A},x_{T}\right)}^{\left(0\right)}\rangle$.

The parameter formation process of the initializing phase is shown in Figure 2.

### 3.2. Signing Phase

**Step S1:** Alice obtains a qubit string |Γ〉 based on the signature information *m*. Suppose there are *n* qubits in the information qubit string $|\Gamma \rangle =\{|\gamma_{1}\rangle ,|\gamma_{2}\rangle$, ⋯, $|\gamma_{n}\rangle \}$, where the symbol {⋯} represents the collection and $|\gamma_{i}\rangle$ represents a single qubit in |Γ〉. Any qubit $|\gamma_{i}\rangle$$\left(i=1,2,\cdots ,n\right)$ in |Γ〉 can be represented as a superposition of two eigenstates |0〉 and |1〉, namely, $|\gamma_{i}\rangle =\alpha_{i}|0\rangle +\beta_{i}|1\rangle$, where $\alpha_{i},\beta_{i}$ are complex numbers that satisfy $|\alpha_{i}{|}^{2}+{\left|\beta_{i}\right|}^{2}=1$. Thus, the signed quantum information string of Alice can be represented as $|\Gamma \rangle =\{\alpha_{1}|0\rangle +\beta_{1}|1\rangle ,\alpha_{2}|0\rangle +\beta_{2}|1\rangle ,\cdots ,\alpha_{n}|0\rangle +\beta_{n}\left|1\rangle \right\}$. Note that if the signature quantum state is known, any copies of |Γ〉 can be prepared in advance. If the signature quantum state is unknown, at least three copies of |Γ〉 are necessary, among which one is combined with 5-particle entangled state, one produces a secret qubit string $|R_{A}\rangle$, and the other is sent to Bob.

**Step S2:** Alice transforms the information qubit string |Γ〉 into a secret qubit string $|R_{A}\rangle =M_{K_{A}}\left(|\Gamma \rangle \right)$ in terms of the secret key $K_{A}$. This transform method can be seen in [14].

**Step S3:** Alice prepares 5-particle entangled states. Alice combines each information qubit with 5-particle entangled state into the same long 6-particle qubit string. Each combinatorial state includes one information particle and five entangled particle. This 6-particle combination state can be described as follows:$$\begin{array}{cc} |\Psi^{i}\rangle_{M12345}\; & =\;|\gamma_{i}\rangle_{M}⊗{|\xi \rangle }_{12345}={\left(\alpha_{i}|0\rangle +\beta_{i}|1\rangle \right)}_{M}⊗{|\xi \rangle }_{12345}\\ & =\;\frac{1}{2\sqrt{2}}[\alpha_{i}|000100\rangle -\alpha_{i}|000111\rangle +\alpha_{i}|001001\rangle -\alpha_{i}|001010\rangle +\alpha_{i}|010000\rangle \\ & +\;\alpha_{i}|010011\rangle +\alpha_{i}|011101\rangle +\alpha_{i}|011110\rangle +\beta_{i}|100100\rangle -\beta_{i}|100111\rangle \\ & +\;\beta_{i}|101001\rangle -\beta_{i}|101010\rangle +\beta_{i}|110000\rangle +\beta_{i}|110011\rangle +\beta_{i}|111101\rangle \\ & +\;\beta_{i}{\left|111110\rangle \right]}_{M12345} \end{array}$$

**Step S4:** Alice uses $\Omega_{A}$ to represent the sequence of *n* (*M*, 2, 3) particles, where M represents the information particle to be signed. $\Omega_{T}$ represents the sequence of *n* (1, 4) particles, and $\Omega_{B}$ represents the sequence of *n* (5) particles. The decoy particles ${|\mu \rangle }_{A,T}=U_{F\left(x_{A},x_{T}\right)}|\varphi_{0}^{\left(0\right)}\rangle =|\varphi_{F\left(x_{A},x_{T}\right)}^{\left(0\right)}\rangle$ and ${|\mu \rangle }_{A,B}=U_{F\left(x_{A},x_{B}\right)}|\varphi_{0}^{\left(0\right)}\rangle =|\varphi_{F\left(x_{A},x_{B}\right)}^{\left(0\right)}\rangle$ are randomly inserted in $\Omega_{T}$ and $\Omega_{B}$ to form $\Omega_{T}^{{}^{\prime }}$ and $\Omega {{}^{\prime }}_{B}$, respectively. Alice sends $\Omega_{T}^{{}^{\prime }}$ to Trent and $\Omega {{}^{\prime }}_{B}$ to Bob.

**Step S5:** Alice performs von Neumann measurement on the particle sequence $\Omega_{A}$ that she has mastered. Suppose the *n*-group von Neumann measurement results are $\delta \left(\Omega_{A}\right)=\left\{\delta \left(\Omega_{A,1}\right),\delta \left(\Omega_{A,2}\right),\cdots ,\delta \left(\Omega_{A,n}\right)\right\}$, where $\Omega_{A,i}\in \left\{\chi^{1},\chi^{2},\cdots ,\chi^{8}\right\}$. Alice encrypts $|R_{A}\rangle$ and $\delta \left(\Omega_{A}\right)$ to form the signature $|S\rangle =E_{K_{A}}\left(|R_{A}\rangle ,\delta \left(\Omega_{A}\right)\right)$ by using quantum one-time pad algorithm [41]. Note that $\delta \left(\Omega_{A}\right)$, even if sometimes described as classical bits, can be converted to qubits from the measurement basis $\left\{\chi^{1},\chi^{2},\cdots ,\chi^{8}\right\}$. Alice sends the signature |S〉 and 2 information qubit strings |Γ〉 to Bob.

### 3.3. Verification Phase

**Step V1:** After confirming that Bob received $\Omega {{}^{\prime }}_{B}$, Alice tells Bob the position of the decoy particles and Bob executes the unitary operation $U_{-F\left(x_{B},x_{A}\right)}$ on the decoy particle ${|\mu \rangle }_{A,B}$, that is, ${|\mu \rangle }_{B,A}=U_{-F\left(x_{B},x_{A}\right)}{|\mu \rangle }_{A,B}$. Then, Bob measures the decoy particles using measurement basis $\{|\varphi_{l}^{\left(0\right)}\rangle$ |l∈q}. If ${|\mu \rangle }_{B,A}\neq |\varphi_{0}^{\left(0\right)}\rangle$, it implies that the identity authentication between Alice and Bob cannot be passed or the decoy particle have been eavesdropped. Finally, Bob calculates the error rate based on measurement outcomes of the decoy particles. If the error rate is less than the previously given value, they perform the next step. Otherwise, the execution of the protocol is aborted. After Bob passes the eavesdropping detection and identity authentication of $\Omega {{}^{\prime }}_{B}$, the decoy particles are removed and $\Omega_{B}$ is restored. Similarly, after confirming that Trent received $\Omega {{}^{\prime }}_{T}$, Alice tells Trent the position of the decoy particles and then Trent executes the unitary operations $U_{-F\left(x_{T},x_{A}\right)}$ on the decoy particle ${|\mu \rangle }_{A,T}$, that is, ${|\mu \rangle }_{T,A}=U_{-F\left(x_{T},x_{A}\right)}{|\mu \rangle }_{A,T}$. Then Trent measures the decoy particles using the measurement basis $\left\{|\varphi_{l}^{\left(0\right)}\rangle |l\in q\right\}$. If ${|\mu \rangle }_{T,A}\neq |\varphi_{0}^{\left(0\right)}\rangle$, it indicates that the identity authentication between Alice and Trent cannot be passed or that the particles are eavesdropped. Finally, Trent calculates the error rate based on measurement outcomes of the decoy particles. If the error rate is less than the previously given value, they perform the next step; otherwise, they abandon the agreement. After Trent performs the eavesdropping detection and identity authentication on $\Omega {{}^{\prime }}_{T}$, the decoy particles are removed and $\Omega_{T}$ is restored.

**Step V2:** After Bob receives |S〉 which was sent by Alice, he encrypts |S〉 and |Γ〉 with the secret key $K_{B}$ to obtain $Y_{B}=E_{K_{B}}\left(|S\rangle ,|\Gamma \rangle \right)$. Bob sends $Y_{B}$ to Trent via a quantum channel.

**Step V3:** After receiving $Y_{B}=E_{K_{B}}\left(|S\rangle ,|\Gamma \rangle \right)$, Trent decrypts it using secret key $K_{B}$ to obtain |S〉 and |Γ〉, and decrypts |S〉 using secret key $K_{A}$ to obtain $|R_{A}\rangle$ and $\delta \left(\Omega_{A}\right)$. In the meantime, Trent measures $\Omega_{T}$ with measurement basis $\{|\phi^{+}\rangle ,|\phi^{-}\rangle ,|\psi^{+}\rangle ,|\psi^{-}\rangle \}$ to obtain the measurement outcome $\delta \left(\Omega_{T}\right)$. Trent uses the secret key $K_{A}$ to transform the information qubit string |Γ〉 into $|R_{A}^{{}^{\prime }}\rangle$ and compare $|R_{A}\rangle$ with $|R_{A}^{{}^{\prime }}\rangle$. If $|R_{A}\rangle =|R_{A}^{{}^{\prime }}\rangle$, Trent sets the initial check parameter θ=1; otherwise, he sets θ=0. Note that this step and the subsequent comparison of the quantum states can be found in [14,42]. To ensure the integrity of the signature, Trent selects an appropriate hash function $H\left(.\right)$ and calculates $H\left(|S\rangle \right)$.

**Step V4:** Trent encryptions |S〉, $H\left(|S\rangle \right)$, $\delta \left(\Omega_{A}\right)$, $\delta \left(\Omega_{T}\right)$, θ with secret key $K_{B}$ and sends $Y_{TB}=E_{K_{B}}(|S\rangle$, $H\left(|S\rangle \right),\delta \left(\Omega_{A}\right),\delta \left(\Omega_{T}\right),\theta )$ to Bob.

**Step V5:** Bob decrypts $Y_{TB}$ to obtain |S〉, H(|S〉), $\delta (\Omega_{A})$, $\delta (\Omega_{T})$ and θ. If θ=0, Bob can assume that the signature was forged, he rejects the signature and exits the verification process; otherwise, Bob continues with the next verification process.

**Step V6:** According to the values of $\delta \left(\Omega_{A}\right)$ and $\delta \left(\Omega_{T}\right)$, Bob chooses the corresponding unitary operator $U_{\left(5\right)}$ in Table 4. Bob performs unitary operation $U_{\left(5\right)}$ on the particles in sequence $\Omega_{B}$ and measures them to obtain the quantum state $|\Gamma^{{}^{\prime }}\rangle$. Notice that $|\Gamma^{{}^{\prime }}\rangle$ is the result of executing controlled quantum teleportation. Then, he compares whether it is equal to |Γ〉. If $\left|\Gamma \rangle \neq \right|\Gamma^{{}^{\prime }}\rangle$, Bob considers the signature invalid and rejects it. If $\left|\Gamma \rangle =\right|\Gamma^{{}^{\prime }}\rangle$, Bob calculates $H^{{}^{\prime }}\left(|S\rangle \right)$ with the same hash function and compares $H^{{}^{\prime }}\left(|S\rangle \right)$ with $H\left(|S\rangle \right)$. If $H^{{}^{\prime }}\left(|S\rangle \right)=H\left(|S\rangle \right)$, Bob accepts |S〉 as the signature of |Γ〉 from Alice; otherwise, the signature is rejected.

The schematic diagram of the main steps of the arbitrated quantum signature scheme is shown in Figure 3.

## 4. Verifiability Analysis

We can prove that, in this scheme, identity authentication and eavesdropping detection can be conducted between Alice and Bob as well as between Alice and Trent according to the measurement outcomes of the decoy particles. An example for the proposed verifiable arbitrated quantum signature scheme can be seen in Appendix A.

In steps I3 and I4, according to Alice’s share polynomial $F(x_{A},y)$ and Bob’s publicly identified information $x_{B}$, Alice calculates $F(x_{A},x_{B})$ and creates decoy particles ${|\mu \rangle }_{A,B}=U_{F(x_{A},x_{B})}|\varphi_{0}^{\left(0\right)}\rangle =|\varphi_{F(x_{A},x_{B})}^{\left(0\right)}\rangle .$ According to Bob’s share polynomial $F(x_{B},y)$ and Alice’s publicly identified information $x_{A}$, Bob calculates $F(x_{B},x_{A})$. In step V1, after Bob receives ${|\mu \rangle }_{A,B}$, he performs the unitary operation $U_{-F(x_{B},x_{A})}$ on ${|\mu \rangle }_{A,B}$, that is ${|\mu \rangle }_{B,A}=U_{-F(x_{B},x_{A})}{|\mu \rangle }_{A,B}$. According to the properties of symmetric binary polynomials, we have $F\left(x_{B},x_{A}\right)=F\left(x_{A},x_{B}\right)$ and ${|\mu \rangle }_{B,A}=U_{-F(x_{B},x_{A})}U_{F(x_{A},x_{B})}|\varphi_{0}^{\left(0\right)}\rangle =|\varphi_{0}^{\left(0\right)}\rangle$. Without external eavesdropping and cheating on either side, Bob’s measurement outcomes of the decoy particles should be $|\varphi_{0}^{\left(0\right)}\rangle$; otherwise, it can be determined that either identity cheating on both sides or external eavesdropping are occurring. Therefore, Alice and Bob can verify whether identity cheating is occurring according to the measurement outcomes of the decoy particles. Similarly, identity verification and eavesdropping detection can also be conducted between Alice and Trent according to the measurement outcomes of the decoy particles.

## 5. Safety Analysis

A secure quantum signature scheme should be of an unforgeable and undeniable property. In other words, it should meet the following requirements: (1) The signature cannot be forged by an attacker (including external adversary Eve and malicious receiver Bob). (2) The signatory Alice cannot disavow the message and signature she sent, and the receiver Bob cannot disavow that he received the signature. (3) That can be arbitrated if the receiver Bob admits the fact of receiving the signature but disavows the integrity of the signature.

### 5.1. Impossibility of Forgery

If the external attacker Eve tries to forge Alice’s signature |S〉 for her own benefit, she should know the key $K_{A}$. However, due to the unconditional security of quantum key distribution [39,40], this is not possible. In addition, the quantum one-time pad protocol [41] is used to improve the security. Therefore, Eve’s forgery is impossible.

If the malicious receiver Bob tries to forge Alice’s signature $|S\rangle =E_{K_{A}}(|R_{A}\rangle ,\delta \left(\Omega_{A}\right))$ for his own benefit, he must also know Alice’s secret key $K_{A}$. However, for the same reason, he cannot obtain any information about the key $K_{A}$. Thus, Bob cannot obtain the correct $|R_{A}\rangle$. Subsequently, the initial check parameter θ used in the verifying phase will not be right, so the arbitrator Trent will discover this forgery. In a worse case, even if key $K_{A}$ is exposed to Eve, she still cannot forge the signature because she cannot create the appropriate $|R_{A}\rangle$ and $\delta (\Omega_{A})$ to associate with the new message. Bob uses the correlation of the Bell state to find this kind of forged file; further verification of $|R_{A}\rangle =|R_{A}^{\prime }\rangle$ cannot be established without the correct $|R_{A}\rangle$. However, if Bob knew the secret key $K_{A}$, forgery would be inevitable.

We can prove that Eve, an external attacker, cannot entangle a decoy particle or an information particle with an auxiliary particle to steal secret information and forge a signature. See Appendix B for details.

### 5.2. Impossibility of Disavowal by the Signatory and the Verifier

A secure quantum signature scheme should have undeniable property. In other words, once the quantum signature is verified as a valid signature, the signatory cannot disavow the fact that the quantum signature is generated by them. The receiver of the signature cannot disavow the fact that he has received the quantum signature.

#### 5.2.1. Impossibility of Disavowal by the Signatory Alice

Suppose Alice tries to disavow the signature |S〉 that she has signed. As shown in Figure 4, after receiving the signature |S〉, Bob cannot decrypt it without the key $K_{A}$. He can only encrypt |S〉 and |Γ〉 to obtain $Y_{B}$ and sends $Y_{B}$ to Trent. After receiving $Y_{B}$, the arbitrator Trent decrypts $Y_{B}=E_{K_{B}}(|S\rangle ,|\Gamma \rangle )$ and $|S\rangle =E_{K_{A}}(|R_{A}\rangle ,\delta \left(\Omega_{A}\right))$ with $K_{A}$ and $K_{B}$. As the signature $|S\rangle =E_{K_{A}}(|R_{A}\rangle ,\delta \left(\Omega_{A}\right))$ contains the key $K_{A}$ shared only by Alice and Trent, Trent can accurately confirm that the signature |S〉 was signed by Alice. Whether |S〉 is the signature of the message |Γ〉 is determined by the initial check parameter θ calculated by the arbitrator Trent. Because $|R_{A}\rangle =M_{K_{A}}(|\Gamma \rangle )$, $|R_{A}^{\prime }\rangle =M_{K_{A}}(|\Gamma^{\prime }\rangle )$, if $|R_{A}\rangle =|R_{A}^{\prime }\rangle$, namely θ=1, then the signature |S〉 was signed by Alice for the message |Γ〉.

#### 5.2.2. Impossibility of Disavowal by the Verifier Bob

Similarly, as long as Trent receives the $Y_{B}$ sent from Bob, because $Y_{B}=E_{K_{B}}(|S\rangle ,|\Gamma \rangle )$ contains the key $K_{B}$ shared only by Bob and Trent, Trent can confirm that Bob received the signature and cannot change it, that is, Bob cannot disavow the fact that he received the signature. If Alice changes signature |S〉 to $|S^{\prime }\rangle$, her behavior will be found when Bob calculates hash value $H^{\prime }(|S\rangle )$ and compares it with H(|S〉). If Bob admits to receiving the signature, but disavows the integrity of the signature, it can be arbitrated according to the hash value H(|S〉) of |S〉.

In this scheme, the eavesdropping detection also functions as identity authentication, which can strengthen the undeniable property of Alice and Bob. In conclusion, our verifiable arbitrated quantum signature scheme has undeniable security.

## 6. Conclusions

In this paper, we proposed a verifiable arbitrated quantum signature scheme based on five-qubit entangled state. The proposed scheme uses mutually unbiased bases particles as decoy particles, and performs unitary operations on these decoy particles using the function values of symmetric binary polynomials, which can carry out not only eavesdropping detection, but also identity authentication among participants.

Due to the unconditional security of quantum key distribution and the quantum one-time pad, the external attacker Eve cannot know Alice’s key $K_{A}$; she cannot forge Alice’s signature |S〉 for her own benefit. For the same reason, Bob cannot forge Alice’s signature |S〉, either. In order to avoid Alice’s disavowal, we set that when Trent receives Alice’s signature |S〉, the hash function value H(|S〉) of the signature is calculated to ensure the integrity of the signature. After Trent receives $Y_{B}$ and decrypts $Y_{B}$ and $|S\rangle =E_{K_{A}}(|R_{A}\rangle ,\delta \left(\Omega_{A}\right))$, the initial check parameter θ confirms that |S〉 is jointly generated by |Γ〉 and $K_{A}$, which proves that Alice did not cheat. At this time, since Trent had no information on parameter $\Omega_{B}$, he could not forge a new signature. After Bob receives $Y_{TB}=E_{K_{B}}(|S\rangle ,H(|S\rangle ),\delta \left(\Omega_{A}\right),\delta \left(\Omega_{T}\right),\theta )$ and decrypts it, as the information of $\delta \left(\Omega_{A}\right),\delta \left(\Omega_{T}\right)$ and $\Omega_{B}$ are in his grasp at this time, he can use the function of quantum teleportation to reconstruct the information qubit |Γ〉 to judge whether to accept the quantum signature |S〉 signed by Alice.

Different from the signature scheme in classical cryptography, the security of our scheme is guaranteed by the quantum one-time pad [41] and quantum key distribution [39,40]. Therefore, it is unconditionally secure. The five-qubit entangled state plays a key role in quantum information processing tasks and it is the threshold number of qubits required for quantum error correction [43]. The principle of five-photon entanglement and open teleportation was reported in [44] and proved that von Neumann measurement, Bell measurement, and single-particle measurement are all feasible under the current technical and experimental conditions, so the scheme has good application value. Compared with the existing arbitrated quantum signature scheme [10,13,14,17,27], our scheme has high stability and can avoid being disavowed for the integrality of signature |S〉. But due to the large number of qubits used in the scheme, it also experiences the problem of low quantum efficiency.

## Figures and Tables

**Figure 1 entropy-24-00111-f001:**
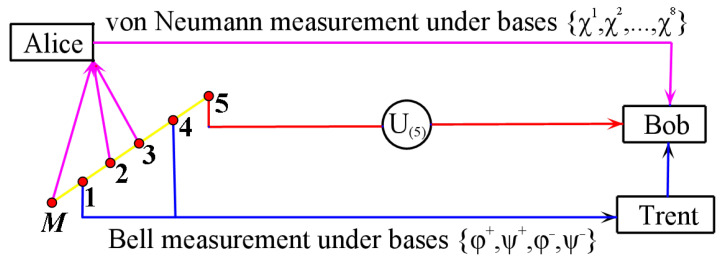
The model of controlled quantum teleportation.

**Figure 2 entropy-24-00111-f002:**
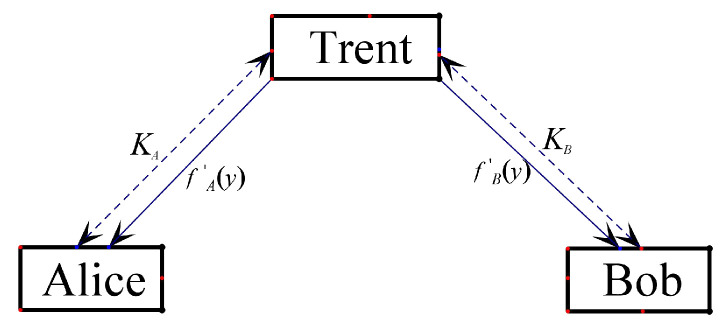
Initializing phase schematic diagram.

**Figure 3 entropy-24-00111-f003:**
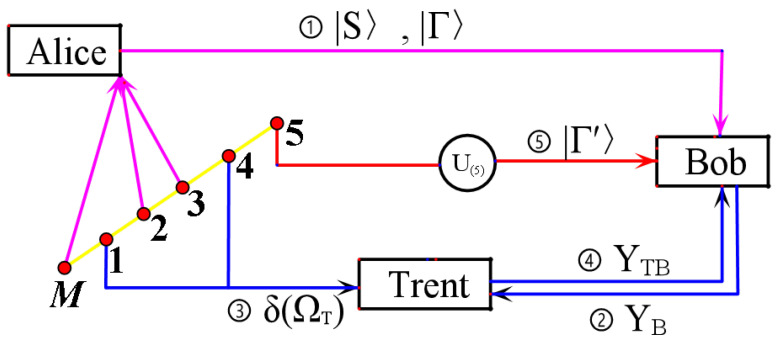
Schematic diagram of the main steps of the arbitrated quantum signature scheme.

**Figure 4 entropy-24-00111-f004:**
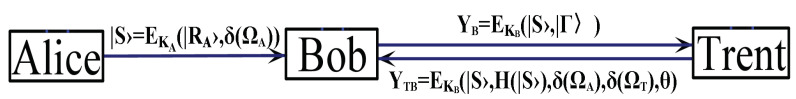
Diagram of transferring signature information.

**Table 1 entropy-24-00111-t001:** The three-particle von Neumann measurement basis.

$\chi^{1}=\frac{1}{\sqrt{2}}\left(|000\rangle +|111\rangle \right)$	$\chi^{2}=\frac{1}{\sqrt{2}}\left(|000\rangle -|111\rangle \right)$
$\chi^{3}=\frac{1}{\sqrt{2}}\left(|001\rangle +|110\rangle \right)$	$\chi^{4}=\frac{1}{\sqrt{2}}\left(|001\rangle -|110\rangle \right)$
$\chi^{5}=\frac{1}{\sqrt{2}}\left(|010\rangle +|101\rangle \right)$	$\chi^{6}=\frac{1}{\sqrt{2}}\left(|010\rangle -|101\rangle \right)$
$\chi^{7}=\frac{1}{\sqrt{2}}\left(|100\rangle +|011\rangle \right)$	$\chi^{8}=\frac{1}{\sqrt{2}}\left(|100\rangle -|011\rangle \right)$

**Table 2 entropy-24-00111-t002:** The outcomes of Alice’s measuring ${|\Psi \rangle }_{M12345}$ with measurement basis $\left\{|\chi^{i}\rangle \right\}$.

${\langle \chi_{M23}^{1}|\Psi \rangle }_{M12345}=\frac{1}{4}\left(\alpha |100\rangle +\alpha |111\rangle +\beta |101\rangle +\beta |110\rangle \right)$	${\langle \chi_{M23}^{2}|\Psi \rangle }_{M12345}=\frac{1}{4}\left(\alpha |100\rangle +\alpha |111\rangle -\beta |101\rangle -\beta |110\rangle \right)$
${\langle \chi_{M23}^{3}|\Psi \rangle }_{M12345}=\frac{1}{4}\left(\alpha |000\rangle -\alpha |011\rangle +\beta |001\rangle -\beta |010\rangle \right)$	${\langle \chi_{M23}^{4}|\Psi \rangle }_{M12345}=\frac{1}{4}\left(\alpha |000\rangle -\alpha |011\rangle -\beta |001\rangle +\beta |010\rangle \right)$
${\langle \chi_{M23}^{5}|\Psi \rangle }_{M12345}=\frac{1}{4}\left(\alpha |001\rangle -\alpha |010\rangle +\beta |000\rangle -\beta |011\rangle \right)$	${\langle \chi_{M23}^{6}|\Psi \rangle }_{M12345}=\frac{1}{4}\left(\alpha |001\rangle -\alpha |010\rangle -\beta |000\rangle +\beta |011\rangle \right)$
${\langle \chi_{M23}^{7}|\Psi \rangle }_{M12345}=\frac{1}{4}\left(\alpha |101\rangle +\alpha |110\rangle +\beta |100\rangle +\beta |111\rangle \right)$	${\langle \chi_{M23}^{8}|\Psi \rangle }_{M12345}=\frac{1}{4}\left(-\alpha |101\rangle -\alpha |110\rangle +\beta |100\rangle +\beta |111\rangle \right)$

**Table 3 entropy-24-00111-t003:** Outcomes of Trent’s measuring ${\langle \chi_{M23}^{i}|\Psi \rangle }_{M12345}$ with Bell measurement basis.

	$|\phi_{14}^{+}\rangle$	$|\phi_{14}^{-}\rangle$	$|\psi_{14}^{+}\rangle$	$|\psi_{14}^{-}\rangle$
${\langle \chi_{M23}^{1}|\Psi \rangle }_{M12345}$	${\left(\alpha |1\rangle +\beta |0\rangle \right)}_{5}$	${\left(-\alpha |1\rangle -\beta |0\rangle \right)}_{5}$	${\left(\alpha |0\rangle +\beta |1\rangle \right)}_{5}$	${\left(-\alpha |0\rangle -\beta |1\rangle \right)}_{5}$
${\langle \chi_{M23}^{2}|\Psi \rangle }_{M12345}$	${\left(\alpha |1\rangle -\beta |0\rangle \right)}_{5}$	${\left(-\alpha |1\rangle +\beta |0\rangle \right)}_{5}$	${\left(\alpha |0\rangle -\beta |1\rangle \right)}_{5}$	${\left(-\alpha |0\rangle +\beta |1\rangle \right)}_{5}$
${\langle \chi_{M23}^{3}|\Psi \rangle }_{M12345}$	${\left(\alpha |0\rangle +\beta |1\rangle \right)}_{5}$	${\left(\alpha |0\rangle +\beta |1\rangle \right)}_{5}$	${\left(-\alpha |1\rangle -\beta |0\rangle \right)}_{5}$	${\left(-\alpha |1\rangle -\beta |0\rangle \right)}_{5}$
${\langle \chi_{M23}^{4}|\Psi \rangle }_{M12345}$	${\left(\alpha |0\rangle -\beta |1\rangle \right)}_{5}$	${\left(\alpha |0\rangle -\beta |1\rangle \right)}_{5}$	${\left(-\alpha |1\rangle +\beta |0\rangle \right)}_{5}$	${\left(-\alpha |1\rangle +\beta |0\rangle \right)}_{5}$
${\langle \chi_{M23}^{5}|\Psi \rangle }_{M12345}$	${\left(\alpha |1\rangle +\beta |0\rangle \right)}_{5}$	${\left(\alpha |1\rangle +\beta |0\rangle \right)}_{5}$	${\left(-\alpha |0\rangle -\beta |1\rangle \right)}_{5}$	${\left(-\alpha |0\rangle -\beta |1\rangle \right)}_{5}$
${\langle \chi_{M23}^{6}|\Psi \rangle }_{M12345}$	${\left(\alpha |1\rangle -\beta |0\rangle \right)}_{5}$	${\left(\alpha |1\rangle -\beta |0\rangle \right)}_{5}$	${\left(-\alpha |0\rangle +\beta |1\rangle \right)}_{5}$	${\left(-\alpha |0\rangle +\beta |1\rangle \right)}_{5}$
${\langle \chi_{M23}^{7}|\Psi \rangle }_{M12345}$	${\left(\alpha |0\rangle +\beta |1\rangle \right)}_{5}$	${\left(-\alpha |0\rangle -\beta |1\rangle \right)}_{5}$	${\left(\alpha |1\rangle +\beta |0\rangle \right)}_{5}$	${\left(-\alpha |1\rangle -\beta |0\rangle \right)}_{5}$
${\langle \chi_{M23}^{8}|\Psi \rangle }_{M12345}$	${\left(-\alpha |0\rangle +\beta |1\rangle \right)}_{5}$	${\left(\alpha |0\rangle -\beta |1\rangle \right)}_{5}$	${\left(-\alpha |1\rangle +\beta |0\rangle \right)}_{5}$	${\left(\alpha |1\rangle -\beta |0\rangle \right)}_{5}$

**Table 4 entropy-24-00111-t004:** The relationship between Alice’s, Trent’s measurement outcomes, and Bob’s unitary operation.

Alice’ MO	Trent’ MO	Bob’s State	$U_{\left(5\right)}$	Trent’ MO	Bob’s State	$U_{\left(5\right)}$
$\chi^{1}=\frac{1}{\sqrt{2}}\left(|000\rangle +|111\rangle \right)$	$|\phi_{14}^{+}\rangle$	${\left(\alpha |1\rangle +\beta |0\rangle \right)}_{5}$	${\left(\sigma_{x}\right)}_{5}$	$|\psi_{14}^{+}\rangle$	${\left(\alpha |0\rangle +\beta |1\rangle \right)}_{5}$	$I_{5}$
	$|\phi_{14}^{-}\rangle$	${\left(-\alpha |1\rangle -\beta |0\rangle \right)}_{5}$	${\left(-\sigma_{x}\right)}_{5}$	$|\psi_{14}^{-}\rangle$	${\left(-\alpha |0\rangle -\beta |1\rangle \right)}_{5}$	$-I_{5}$
$\chi^{2}=\frac{1}{\sqrt{2}}\left(|000\rangle -|111\rangle \right)$	$|\phi_{14}^{+}\rangle$	${\left(\alpha |1\rangle -\beta |0\rangle \right)}_{5}$	${\left(i\sigma_{y}\right)}_{5}$	$|\psi_{14}^{+}\rangle$	${\left(\alpha |0\rangle -\beta |1\rangle \right)}_{5}$	${\left(\sigma_{z}\right)}_{5}$
	$|\phi_{14}^{-}\rangle$	${\left(-\alpha |1\rangle +\beta |0\rangle \right)}_{5}$	${\left(-i\sigma_{y}\right)}_{5}$	$|\psi_{14}^{-}\rangle$	${\left(-\alpha |0\rangle +\beta |1\rangle \right)}_{5}$	${\left(-\sigma_{z}\right)}_{5}$
$\chi^{3}=\frac{1}{\sqrt{2}}\left(|001\rangle +|110\rangle \right)$	$|\phi_{14}^{+}\rangle$	${\left(\alpha |0\rangle +\beta |1\rangle \right)}_{5}$	$I_{5}$	$|\psi_{14}^{+}\rangle$	${\left(-\alpha |1\rangle -\beta |0\rangle \right)}_{5}$	${\left(-\sigma_{x}\right)}_{5}$
	$|\phi_{14}^{-}\rangle$	${\left(\alpha |0\rangle +\beta |1\rangle \right)}_{5}$	$I_{5}$	$|\psi_{14}^{-}\rangle$	${\left(-\alpha |1\rangle -\beta |0\rangle \right)}_{5}$	${\left(-\sigma_{x}\right)}_{5}$
$\chi^{4}=\frac{1}{\sqrt{2}}\left(|001\rangle -|110\rangle \right)$	$|\phi_{14}^{+}\rangle$	${\left(\alpha |0\rangle -\beta |1\rangle \right)}_{5}$	${\left(\sigma_{z}\right)}_{5}$	$|\psi_{14}^{+}\rangle$	${\left(-\alpha |1\rangle +\beta |0\rangle \right)}_{5}$	${\left(-i\sigma_{y}\right)}_{5}$
	$|\phi_{14}^{-}\rangle$	${\left(\alpha |0\rangle -\beta |1\rangle \right)}_{5}$	${\left(\sigma_{z}\right)}_{5}$	$|\psi_{14}^{-}\rangle$	${\left(-\alpha |1\rangle +\beta |0\rangle \right)}_{5}$	${\left(-i\sigma_{y}\right)}_{5}$
$\chi^{5}=\frac{1}{\sqrt{2}}\left(|010\rangle +|101\rangle \right)$	$|\phi_{14}^{+}\rangle$	${\left(\alpha |1\rangle +\beta |0\rangle \right)}_{5}$	${\left(\sigma_{x}\right)}_{5}$	$|\psi_{14}^{+}\rangle$	${\left(-\alpha |0\rangle -\beta |1\rangle \right)}_{5}$	$-I_{5}$
	$|\phi_{14}^{-}\rangle$	${\left(\alpha |1\rangle +\beta |0\rangle \right)}_{5}$	${\left(\sigma_{x}\right)}_{5}$	$|\psi_{14}^{-}\rangle$	${\left(-\alpha |0\rangle -\beta |1\rangle \right)}_{5}$	$-I_{5}$
$\chi^{6}=\frac{1}{\sqrt{2}}\left(|010\rangle -|101\rangle \right)$	$|\phi_{14}^{+}\rangle$	${\left(\alpha |1\rangle -\beta |0\rangle \right)}_{5}$	${\left(i\sigma_{y}\right)}_{5}$	$|\psi_{14}^{+}\rangle$	${\left(-\alpha |0\rangle +\beta |1\rangle \right)}_{5}$	${\left(-\sigma_{z}\right)}_{5}$
	$|\phi_{14}^{-}\rangle$	${\left(\alpha |1\rangle -\beta |0\rangle \right)}_{5}$	${\left(i\sigma_{y}\right)}_{5}$	$|\psi_{14}^{-}\rangle$	${\left(-\alpha |0\rangle +\beta |1\rangle \right)}_{5}$	${\left(-\sigma_{z}\right)}_{5}$
$\chi^{7}=\frac{1}{\sqrt{2}}\left(|100\rangle +|011\rangle \right)$	$|\phi_{14}^{+}\rangle$	${\left(\alpha |0\rangle +\beta |1\rangle \right)}_{5}$	$I_{5}$	$|\psi_{14}^{+}\rangle$	${\left(\alpha |1\rangle +\beta |0\rangle \right)}_{5}$	${\left(\sigma_{x}\right)}_{5}$
	$|\phi_{14}^{-}\rangle$	${\left(-\alpha |0\rangle -\beta |1\rangle \right)}_{5}$	$-I_{5}$	$|\psi_{14}^{-}\rangle$	${\left(-\alpha |1\rangle -\beta |0\rangle \right)}_{5}$	${\left(-\sigma_{x}\right)}_{5}$
$\chi^{8}=\frac{1}{\sqrt{2}}\left(|100\rangle -|011\rangle \right)$	$|\phi_{14}^{+}\rangle$	${\left(-\alpha |0\rangle +\beta |1\rangle \right)}_{5}$	${\left(-\sigma_{z}\right)}_{5}$	$|\psi_{14}^{+}\rangle$	${\left(-\alpha |1\rangle +\beta |0\rangle \right)}_{5}$	${\left(-i\sigma_{y}\right)}_{5}$
	$|\phi_{14}^{-}\rangle$	${\left(\alpha |0\rangle -\beta |1\rangle \right)}_{5}$	${\left(\sigma_{z}\right)}_{5}$	$|\psi_{14}^{-}\rangle$	${\left(\alpha |1\rangle -\beta |0\rangle \right)}_{5}$	${\left(i\sigma_{y}\right)}_{5}$

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
