# Peer review of "A Verifiable Arbitrated Quantum Signature Scheme Based on Controlled Quantum Teleportation"

_entropy, 2022, doi:10.3390/e24010111_

Round 1
Reviewer 1 Report
The paper describes in detail a protocol of arbitrated quantum signature that makes use of quantum teleportation. The scheme is clearly presented and, in the introduction, the authors made a successful effort to contextualize the work within the literature on the subject. I think this work should be published because it presents a valuable addition to the topic.
I recommend nevertheless that the authors review the manuscript carefully due to some typos. It must be mentioned that the copy made available to me had already some blue markings indicating a previous revision, but even so, some typos remain and improvements in redaction should be considered. Other than that I think the paper is suitable for publication.
Reviewer 2 Report
I think that the contribution of this paper is quite straightforward as they introduce a rather complex and technical verifiable arbitrated quantum signature scheme based on controlled quantum teleportation.
They tackle what is, in my opinion, a very interesting and promising line of research. The organization of the paper is very good and it this makes it easy, even for non-experts, to follow their rationale and their conclusions. The introduction and related work sections are truly great and indicate the expertise of the authors in this field. The equations, the Tables and the Figures facilitate the understanding of the paper. It is quite evident that the authors have put a lot of effort in this work. Finally, the Appendix is very helpful in understanding their protocol.
To be honest I have to admit that this paper, being very technical and very equation oriented, is very difficult for the reviewer to verify the correctness of all the equations. I have done my best to check them, and they indeed seem to be correct, but, still, I cannot be 100% sure that all mathematical mistakes or typos have been eliminated.
I have detected just a few typos and awkward English expressions:
- Page 2, line 85: the year should be 1993 instead of 1983
- Page 4, lines 147-148: the phrase ”Additionally, this can be found at most“ could be better phrased
- Page 4, line 179: I believe it should be “i = 1, 2, ..., 8” instead of “i = 1, 2, ..., 16“
- Page 7, line 246: I believe it should be “that satisfy” instead of “that satisfies”
- Page 8, line 247: is there a typo in “β_2i |12>” ?
- Page 12, line 423: Perhaps the title “An example for the proposed scheme” would be more fitting?
My final impression is that this paper is very interesting, very well organized and presented. The authors must have done a lot of work for their proposed scheme. Thus, I think that it can be accepted for publication as is or after very minor revisions.
Author Response
Please see the attachment

This manuscript is a resubmission of an earlier submission. The following is a list of the peer review reports and author responses from that submission.
Round 1
Reviewer 1 Report
Please see attached file for my report.

Author Response
Response:
We have tried to improve the presentation to make it more readable as much as possible. We requested the English editing service and our paper has undergone English language editing by MDPI. The grammar and common technical terms in our paper are thought to be correctly used and it is edited to a level suitable for reporting research in a scholarly journal. We have made a few adjustments to the structure of the manuscript, we hope that it meets the requirement.
The security of classical cryptography is mainly based on difficult problems such as large number factorization, discrete logarithms problem and elliptic curve discrete logarithm problem. Most of the schemes do not have unconditional security, but they are relatively mature. However, Shor introduced a polynomial-time algorithms which increasingly makes the security of classical cryptographic protocols fragile[Shor, P.W. Polynomial-Time Algorithms for Prime Factorization and Discrete Logarithms on a Quantum Computer. SIAM J. Comput. 1997, 26, 1484–1509]. Different from the signature scheme in classical cryptography, the security of our scheme is guaranteed by the quantum one-time pad and quantum key distribution, which has been proven to provide unconditionally security. Therefore, our scheme is unconditionally secure.
The five-qubit entangled state plays a key role in quantum information processing tasks and it is the threshold number of qubits required for quantum error correction [Bennett, C.H.; DiVincenzo, D.P.; Smolin, J.A.; Wootters W.K. Mixed-state entanglement and quantum error correction. Phys. Rev. A 1996, 54, 3824]. The principle of five-photon entanglement and open teleportation were reported in the literature. It proved that von Neumann measurement, Bell measurement and single-particle measurement are all feasible under the current technical and experimental conditions, so the scheme has good application value[Zhao, Z.; Chen, Y.A.; Zhang, A.N.; Yang, T.; Briegel, H.J.; Pan, J.W. Experimental demonstration of five-photon entanglement and open-destination teleportation. Nature, 2004, 430: 54-58].
Due to the large number of quantum bits used, compared with the existing AQS, our scheme has high stability and can resist integrity denial attack, but it has the problem of low quantum efficiency.

Reviewer 2 Report
1) typo line 229, the number 2 in the ket must be 1
2) line 257-258 which vlaue of error rate is the one you are referring to?
3) line 180 in the ket ther is a 5, it must be a 1
4) is is necessary tocohose the position of the decoy state at random?
5) line 423 which must be changed to as (or with "which is")
6) line 431 clarify the conclusion with a refernce to the reasone why tht must be necessary the case.
7) line 61 chaind must be turned into chained.
Additional comments:
I believe that the approach is interesting because it allows the parties to be sure that now one is cheating (Alice and bob thanks to the presence of Trent and Trent thanks to the QKD link between the two other parties).
From the text in my opinion is not completely clear why the authors chose 5 qubits entangled states, if I understand correctly the robustness of the 5 qubit states comes when 4 qubits are used for redundancy while in this approach it look like only 2 qubits are used this way.
Maybe it would be useful to have a wider explanation.
I was also wondering if instead of using this scheme, which is definitely very inefficient, (the generation and distribution of 5 entangled states is complicated and the rate I believe is low), it would be equivalent but more efficient to use four standard qkd link Alice-Trent, Alice-Bob, Trent-Alice Trent-Bob to fulfill the same logical security.
I also believe that the examples are a little heavy to read during the text because essentially the authors wrote essentially the same sentences as in the description of the protocol adding some explicit calculations, I believe it might be more effective if they put those calculations in a dedicated appendix.
It would also be beneficial to give an explanation on the choice of the assignment of the qubits to each partner, I don't understand if it is the only possible choice or there's room for different possible assignments.
I also believe it would be more clear if the authors explained why they chose a prime number for the modulo of the symmetric polynomial.
It is also not very convincing the security analysis because it seems from the text that Eve can only possibly attack the decoy states, which is not necessarily the case.
Author Response
Point 1: typo line 229, the number 2 in the ket must be 1
Response 1: Thanks. I have corrected it.
Point 2: line 257-258 which value of error rate is the one you are referring to?
Response 2: If the number of is m and the number of decoy particles is n, then the error rate can be expressed as m/n. So we have completed the form of expression: Finally, Bob calculates the error rate based on measurement outcomes of the decoy particles (line 281). Finally, Trent calculates the error rate based on measurement outcomes of the decoy particles (line 291).
Point 3: line 180 in the ket ther is a 5, it must be a 1
Response 3: Thanks. I have corrected it. We also corrected other errors in Table 5.
Point 4: Is it necessary to choose the position of the decoy state at random?
Response 4: Inspired by your comments, we thought that the position of the decoy particles could be fixed without random selection, so we added one to lines 123: Since the location of the decoy particles are fixed, the receiver neither only needs to ask about the location of the decoy particles nor ask what the measurement bases are in the process of performing eavesdropping detection. We have also made the corresponding modifications in Step V1 of verification phase.
Point 5:line 423 which must be changed to as (or with "which is")
Response 5: Thanks. I have corrected it.
Point 6: line 431 clarify the conclusion with a reference to the reason why it must be necessary the case.
Response 6: Our expression is incorrect because we mistakenly used "agreement key". We have replaced “agreement key” with “signing messages”.
Point 7: line 61 chaind must be turned into chained.
Response 7: Thanks. I have corrected it.
Additional comments:
Point 1: I believe that the approach is interesting because it allows the parties to be sure that now one is cheating (Alice and bob thanks to the presence of Trent and Trent thanks to the QKD link between the two other parties).
Response 1: Thanks.
Point 2: From the text in my opinion is not completely clear why the authors chose 5 qubits entangled states, if I understand correctly the robustness of the 5 qubit states comes when 4 qubits are used for redundancy while in this approach it look like only 2 qubits are used this way. Maybe it would be useful to have a wider explanation. I was also wondering if instead of using this scheme, which is definitely very inefficient, (the generation and distribution of 5 entangled states is complicated and the rate I believe is low), it would be equivalent but more efficient to use four standard QKD link Alice-Trent, Alice-Bob, Trent-Alice Trent-Bob to fulfill the same logical security.
Response 2: The five-qubit entangled state plays a key role in quantum information processing tasks and it is the threshold number of qubits required for quantum error correction [43]. The principle of five-photon entanglement and open teleportation was reported in the literature [44] and proved that von Neumann measurement, Bell measurement and single-particle measurement are all feasible under the current technical and experimental conditions, so the scheme has good application value. Five-qubit entangled state can be used to perfect teleportation of arbitrary single- and two-qubit systems [37], which are suitable for maximum contact teleportation and satisfy the biggest task-oriented definition of entangled states [36]. Due to the above advantages, in this paper, we use the five-qubit entangled state as the quantum channel to execute arbitrated quantum signature scheme based on controlled quantum teleportation. Due to the large number of quantum bits used, compared with the existing AQS, our scheme has high stability and can resist integrity denial attack, but also experiences the problem of low quantum efficiency.
Point 3: I also believe that the examples are a little heavy to read during the text because essentially the authors wrote essentially the same sentences as in the description of the protocol adding some explicit calculations, I believe it might be more effective if they put those calculations in a dedicated appendix.
Response 3: Thanks. We have put those calculations in Appendix A.
Point 4: It would also be beneficial to give an explanation on the choice of the assignment of the qubits to each partner, I don't understand if it is the only possible choice or there's room for different possible assignments.
Response 4: The question you ask is an interesting question. In our scheme, the six particles are assigned as follows: (M, 2, 3) to Alice, (1, 4) to Trent, and (5) to Bob. We also tried some other allocation methods, but they were unsuccessful. But your question will lead us to further thinking.
Point 5: I also believe it would be more clear if the authors explained why they chose a prime number for the modulo of the symmetric polynomial.
Response 5: When q is a prime number, Fq is a finite field and GF(q)[x,y] is also a finite field. In addition, if q is a prime number, all elements in the set have additive and multiplicative inversion (except 0). Inspired by your comments, we have added a statement to the text: F(x, y)∈GF(q)[x,y]. (Refer to literature: Liu et. al., cheating identifiable secret sharing scheme using symmetric bivariate polynomial, Information Sciences, 2018, 453: 21–29).
Point 6: It is also not very convincing the security analysis because it seems from the text that Eve can only possibly attack the decoy states, which is not necessarily the case.
Response 6: We improve the safety requirements of quantum signature scheme: (1) the signature cannot be forged by an attacker (including external adversary Eve and malicious receiver Bob); (2) the signer Alice cannot deny the message and signature she sent, and the receiver Bob cannot deny that he received the signature; (3) that can be arbitrated if the receiver Bob admits the fact of receiving the signature but denies the integrity of the signature. We proved that an external attacker Eve, cannot entangle a decoy particle or an information particle with an auxiliary particle to steal secret information and forge a signature, which can be seen in Appendix B for details.

Reviewer 3 Report
The paper presents a communication protocol called arbitrated quantum signature. The protocol is quite involved and difficult to understandas themotivation and scheme only is explained in chapter 3, and is tailored to readers familiar with the concept and purpose of this signature scheme.
As such I do not recommend publication but urge the authors to improve the presentation to make it more accessible to non experts.
It is positive that controlled teleportation is explained, as it might be useful for other stuff, but the transition is not quite clear, as the signature protocol requires quantum communication and manipulations which are not included in controlled quantum teleportation.
The usage of decoy particles is unclear, and I am not sure, whether this already makes the scheme secure against attacks by Eve. The attacks described here are special, and it is not clear to me whether this already would include any general attack. Limiting the capabilities of eavsdropper limits the applicability of the protocol. I would like to propose a more involved security analysis and/or specification.
Author Response
Point 1: The paper presents a communication protocol called arbitrated quantum signature. The protocol is quite involved and difficult to understand as the motivation and scheme only is explained in chapter 3, and is tailored to readers familiar with the concept and purpose of this signature scheme. As such I do not recommend publication but urge the authors to improve the presentation to make it more accessible to non-experts.
Response 1: We have tried to improve the presentation to make it more accessible to non-experts as much as possible. We requested the English editing service and our paper has undergone English language editing by MDPI. The grammar and common technical terms in our paper are thought to be correctly used and it is edited to a level suitable for reporting research in a scholarly journal.
Point 2: It is positive that controlled teleportation is explained, as it might be useful for other stuff, but the transition is not quite clear, as the signature protocol requires quantum communication and manipulations which are not included in controlled quantum teleportation.
Response 2: Figure 1 and Figure 3 can intuitively represent that our arbitrated quantum scheme is constructed based on the controlled quantum teleportation. To emphasize that, we added a statement in line 316: Notice that, is result of controlled quantum teleportation.
Point 3: The usage of decoy particles is unclear, and I am not sure, whether this already makes the scheme secure against attacks by Eve. The attacks described here are special, and it is not clear to me whether this already would include any general attack. Limiting the capabilities of eavesdropper limits the applicability of the protocol. I would like to propose a more involved security analysis and/or specification.
Response 3: We improve the safety requirements of quantum signature scheme: (1) the signature cannot be forged by an attacker (including external adversary Eve and malicious receiver Bob); (2) the signer Alice cannot deny the message and signature she sent, and the receiver Bob cannot deny that he received the signature; (3) that can be arbitrated if the receiver Bob admits the fact of receiving the signature but denies the integrity of the signature. We proved that an external attacker Eve, cannot entangle a decoy particle or an information particle with an auxiliary particle to steal secret information and forge a signature, which can be seen in Appendix B for details.

Reviewer 4 Report
This manuscript designed an arbitrated quantum signature scheme using 5 qubit quantum teleportation. The protocol can ensure the unforgeable and undeniable property of a quantum signature scheme. The article is well written and relatively easy to follow. I would recommend the manuscript for publication with the necessary changes outlined below.
- Does the arbitrator Trent need to be a trusted party? Maybe it is good to state this somewhere in the manuscript.
- In lines 37 and 429, the author mentioned about denial attack. How does the proposed scheme resist denial attacks?
- In line 123, the author mentioned that the location and measurement bases for the decoy particles are fixed. Could Eve know such information? What if somehow she knows the information? Is the analysis in section 4 still hold?
- In Table 4, there seems to be 2 plus signs in χ3.
- In line 227, could the author define the set Fq?
- In line 240, could the author explain how to make sure the decoy particles is “enough”? How is the number of q selected?
- In line 249, what does “plural” mean here?
- In line 251, could the author explain the claim “Note that if the signature quantum state is known, may prepare any copies in advance”? Alice prepared the state by herself. How come the state is not known?
- In steps V3 and V4, how does Trent compare two states RA and RA’? Does he need to know the measurement bases? After Trent compares the states, how he can reconstruct the signature S?
- In line 527, in the 3rd line of proof, the term ωk, ω2k and ωk(q-1) should be ωk, ω2k and ωk(q-1). There might be something wrong in line 528 shown as a box.
- In line 105, there is a typo “sigle”.
Author Response
Thank you for your comments concerning our manuscript entitled "A Verifiable Arbitrated Quantum Signature Scheme Based on Controlled Quantum Teleportation" submitted to the special issue "Practical Quantum Communication" of Entropy. This paper is focused on a verifiable arbitrated quantum signature scheme. Those comments are all valuable and very helpful for revising and improving our paper, as well as the important guiding significance to our researches. We have studied comments carefully and made correction which we hope meet with approval. The details of our revisions and responses to the comments are as follows:
Point 1: Does the arbitrator Trent need to be a trusted party? Maybe it is good to state this somewhere in the manuscript.
Response 1: Thank you very much for your valuable suggestions to our manuscript sincerely. We have added the statement “The arbitrator Trent should be trusted by both Alice and Bob.” in line 218.
Point 2. In lines 37 and 429, the author mentioned about denial attack. How does the proposed scheme resist denial attacks?
Response 2: We introduce undeniable property of our scheme in Section 5.2, it involves that undeniable property of signer Alice in Section 5.2.1. in line 374 and the undeniable property of verifier Bob in Section 5.2.2. in line 385. How the proposed scheme resists denial attacks is explained in this section.
Point 3. In line 123, the author mentioned that the location and measurement bases for the decoy particles are fixed. Could Eve know such information? What if somehow she knows the information? Is the analysis in section 4 still hold?
Response 3: In line 123, we mentioned that the location and measurement bases for the decoy particles are fixed, and our scheme allows Eve to know such information. If Eve knows the location and measurement bases for the decoy particles, he still cannot pass the eavesdropping detection. The reason is that he cannot get the share polynomial fA(y) or fB(y), and he cannot calculate F(xA, xB). Thus the correct unitary operation on the decoy particles cannot be performed. So even if Eve knows the information, the analysis in section 4 is still hold.
Point 4. In Table 4, there seems to be 2 plus signs in χ3.
Response 4: Thanks. We truly appreciate your careful work. We have corrected it.
Point 5. In line 227, could the author define the set Fq?
Response 5: Thank you very much for your guidance to improve our manuscript. The Fq is a finite field in line 227. We have explained it in line 228.
Point 6. In line 240, could the author explain how to make sure the decoy particles is “enough”? How is the number of q selected?
Response 6: To ensure the integrity of |S⟩, it is enough for Alice to prepare 3n decoy particles in the step I4. The unconditional security obtained in quantum cryptographic scheme relies on the fact that any attempt of eavesdropping can be identified. According to literature [19], we always need to check half of the travel qubits to obtain an unconditional security. To meet this requirement, we need to prepare the same number of decoy particles in the step I4 for ΩT and ΩB, which are used in step S4. Since there are 2n information particles in ΩT and n information particles in ΩB, we need prepare at least 3n decoy particles. As can be read in line 142, “additionally, this can be found at most q+1 mutually unbiased bases if q is an odd prime number”. For consistency, we chose q as an odd prime number.
Point 7. In line 249, what does “plural” mean here?
Response 7: We appreciate your careful work earnestly. Here we used the inappropriate word “plural”, it should be replaced by “complex numbers”. We have corrected it.
Point 8. In line 251, could the author explain the claim “Note that if the signature quantum state is known, may prepare any copies in advance”? Alice prepared the state by herself. How come the state is not known?
Response 8: Alice was a signer, but she is not necessarily the provider of signature quantum states. The signature quantum states are provided depending on the specific applications. If Alice knows the signature quantum states, she can prepare many copies for them. If Bob knows it, he can provide many copies of the signature quantum states to Alice. And if the identities of the states are unknown to both Alice and Bob, the third party, who needs Alice’s signature of certain message to be verified by Bob, is required to offer Alice the signature quantum states. To express the above meaning clearly, we replaced “creates” with “obtains” in line 245. Your enlightening question really helps us to revise our manuscript.
Point 9. In steps V3 and V4, how does Trent compare two states RA and RA’? Does he need to know the measurement bases? After Trent compares the states, how he can reconstruct the signature S?
Response 9: Although we do not explain the method of comparing the two quantum states, a pathway for reference is provided in line 305: “Note that this step and the subsequent comparison of the quantum states may be found in the literature [14, 42]”. In step V6, the way of comparing |Γ⟩ and |Γ′⟩ is the same as that of comparing |RA⟩ and |R′A⟩ in Step V3. As shown in literature [14, 42], Trent does not need to know the measurement bases when comparing between the two states. His comparison between them does not destroy the integrity of |R⟩, so |S⟩ can be reconstructed with |R⟩ and δ(ΩA). We really hope the explanation provided here can be approved.
Point 10. In line 527, in the 3rd line of proof, the term ωk, ω2k and ωk(q-1) should be ωk, ω2k and ωk(q-1). There might be something wrong in line 528 shown as a box.
Response 10: Thank you for your valuable suggestion. We have corrected this error pointed out in line 527. This box represents the end of the lemma. We have put it at the end of line 528.
Point 11. In line 105, there is a typo “sigle”.
Response 11: Thank you again for your careful work. We have replaced the wrong spelling word "sigle" with the correct word "single".
Thank you very much for your comments and suggestions. You have given us valuable guidance on the revision of the manuscript.
Yours sincerely,
Dianjun Lu

Round 2
Reviewer 1 Report
The authors have not addressed any of the issues raised in my previous report. Their reply, is very general, and meaningless. As described in my first report, there exists a classical, unconditionally secure digital signature scheme, which is far more flexible and efficient than the authors' scheme, and can cover much larger distances. The authors' work does not offer anything useful, and thus I cannot recommend its publication.
Author Response
It is regretful for not getting your approval, we thank you for your comments. We will make persistent efforts in our future work.
Reviewer 4 Report
The author addressed most of my concerns, but I still need to clarify some minor flaws. I would recommend the manuscript for publication with the necessary changes outlined below.
- In response 2, is it common to use “denial” instead of “disavowal”? I was confused with the denial-of-service attack.
- In response 3, if Eve knows such information, she can ignore the decoy particles and eavesdrop on everything else. If Alice and Bob only check eavesdroppers using the decoy particles, the result can pass.
- In response 9, I understand we can do state comparison using the methods from [14,42], but they are probabilistic. How do the errors affect your protocol security?
Round 3
Reviewer 4 Report
I'm sorry, the authors still do not answer my concerns clearly. The reply and revised version still have serious flaws that should have been considered before the first submission. Since the objective of this paper is to design this protocol, after careful consideration, I have decided that I cannot recommend the publication of this paper. The reasons are listed below.
- In step V1 of appendix, the statement of Alice telling Bob the position should also be restored.
- The entangled states sent to Trent have 2 particles, while the decoy states contain only 1. Does this give Eve the location information of decoy particles? Furthermore, since the decoy-state has a much higher dimension than the information states, is Eve allowed to do any measurement to tell the difference (even only probabilistic measurement)? How will this affect the error rate mentioned in line 482?
- In response 3, the protocol requires Trent to say for sure that the states are the same or different to maintain security, which is impossible. The author claimed in the response that they could compare m times to control the error, but the author also claimed in line 253 that the protocol requires only down to 3 copies of qubit string. Also, if the three parties can control their measurement error by sending multiple copies, Eve can also use them to reduce her measurement error.